# Associations of FGF21 and GDF15 with mitochondrial dysfunction in children living with perinatally-acquired HIV: A cross-sectional evaluation of pediatric AIDS clinical trials group 219/219C

**Greg S. Gojanovich**[1], **Denise L. Jacobson**[2], **Carly Broadwell**[2], **Brad Karalius**[2], **Brian Kirmse**[3], **Mitchell E. Geffner**[4], **Jennifer Jao**[5], **Russell B. Van Dyke**[6], **Elizabeth J. McFarland**[7], **Margarita Silio**[6], **Marilyn Crain**[8], **Mariana Gerschenson**[1]*, **for the Pediatric HIV/AIDS Cohort Study**

1 University of Hawaii at Manoa, Honolulu, Hawaii, United States of America, 2 Harvard TH Chan School of Public Health, Boston, Massachusetts, United States of America, 3 University of Mississippi Medical Center, Jackson, Mississippi, United States of America, 4 Keck School of Medicine of USC, Los Angeles, California, United States of America, 5 Northwestern Feinberg School of Medicine, Chicago, Illinois, United States of America, 6 Tulane University School of Medicine, New Orleans, Louisiana, United States of America, 7 University of Colorado School of Medicine, Aurora, Colorado, United States of America, 8 University of Alabama at Birmingham, Birmingham, Alabama, United States of America

* gerschen@hawaii.edu

**Data Availability Statement:** Data cannot be shared publicly because they may contain

## Abstract

### Background

In persons living with HIV, mitochondrial disease (MD) is difficult to diagnose, as clinical signs are non-specific with inconsistent patterns. Fibroblast growth factor 21 (FGF21) and growth differentiation factor 15 (GDF15) are mitokines elevated in MD patients without HIV, and associated with cardiometabolic comorbidities in adults living with HIV. We assessed relationships of these biomarkers with MD in children living with perinatally-acquired HIV infection (CPHIV).

### Setting

Cross-sectional study of CPHIV from Pediatric ACTG 219/219C classified by Mitochondrial Disease Criteria (MDC) that defines scores 2–4 as "possible" MD.

### Methods

Each case with MDC equaling 4 (MDC4; n = 23) was matched to one randomly selected control displaying no MDC (MDC0; n = 23) based on calendar date. Unmatched cases with MDC equaling 3 (MDC3; n = 71) were also assessed. Plasma samples proximal to diagnoses were assayed by ELISA. Mitokine distributions were compared using Wilcoxon tests, Spearman correlations were calculated, and associations with MD status were assessed by conditional logistic regression.

potentially identifying or sensitive patient information. Data are available from the PHACS Ethics Committee (https://phacsstudy.org/Our-Research/Data-Request-Form) for researchers who meet the criteria for access to confidential data.

**Funding:** This study was supported by the University of Hawaii and the US Department of Health and Human Services, National Institutes of Health grants: P20GM113134 (Gerschenson, M) and U54MD007584 [Hedges, J and Mokuau, N (PI)]. The funders had no role in study design, data collection and analysis, decision to publish, or preparation of the manuscript.

**Competing interests:** MEG has a research contract with NovoNordisk; is a member of advisory boards for Daiichi Sankyo, Ferring, Novo Nordisk, Nutritional & Growth Solutions, Millendo, Pfizer, and Spruce Biosciences; serves on data safety monitoring boards for Ascendis, Millendo, and Tolmar; and receives royalties from UpToDate and McGraw-Hill. MG has been a consultant for Abbott and Oncolys Biopharma. Mr. Karalius reports grants from National Institutes of Health, grants from US Department of Health and Human Services, grants from NICHD, grants from NIDCR, grants from NINDS, grants from NIDCD, grants from NIAID, grants from NIMH, grants from NIDA, grants from NIAAA, grants from NCI, grants from OAR, grants from NHLBI, grants from Harvard TH Chan School of Public Health, grants from Tulane University School of Medicine, during the conduct of the study. Dr. Jao reports grants from National Institutes of Health, during the conduct of the study. Dr. Van Dyke reports grants from National Institutes of Health, during the conduct of the study. Dr. Gerschenson reports grants from National Institutes of Health, grants from US Department of Health and Human Services, grants from NICHD, grants from NIDCR, grants from NINDS, grants from NIDCD, grants from NIAID, grants from NIMH, grants from NIDA, grants from NIAAA, grants from NCI, grants from OAR, grants from NHLBI, grants from Harvard TH Chan School of Public Health, grants from Tulane University School of Medicine, during the conduct of the study.

## Results

Median FGF21 and GDF15 concentrations, respectively, were highest in MDC4 (143.9 and 1441.1 pg/mL), then MDC3 (104.0 and 726.5 pg/mL), and lowest in controls (89.4 and 484.7 pg/mL). Distributions of FGF21 (paired Wilcoxon rank sum p = 0.002) and GDF15 (paired Wilcoxon rank sum p<0.001) differed in MDC4 vs MDC0. Mitokine concentrations were correlated across all participants (r = 0.33; p<0.001). Unadjusted odds ratios of being MDC4 vs MDC0 were 5.2 [95% confidence interval (CI): 1.06–25.92] for FGF21 and 3.5 (95%CI: 1.19–10.25) for GDF15. Relationships persisted after covariate adjustments.

## Conclusion

FGF21 and GDF15 levels may be useful biomarkers to screen for CPHIV with mitochondrial dysfunction.

## Introduction

Mitochondrial dysfunction, without intervention, can result in multi-organ diseases that severely impact quality of life. Increased mitochondrial dysfunction, indicated by reduced mitochondrial DNA (mtDNA) or oxidative phosphorylation (OXPHOS) levels, and inter-related cardiometabolic complications, such as insulin resistance (IR) and cardiovascular disease, have been observed in children living with perinatally-acquired HIV infection (CPHIV). However, biomarkers associated with dysfunction or overt mitochondrial disease (MD), e.g. symptoms meeting clinically-defined Mitochondrial Disease Criteria (MDC) of Wolf and Smeitink [1], in CPHIV have not been validated as done for adults and children living in the absence of HIV infection [2, 3].

Mitokines are soluble factors generated in response to local mitochondrial stressors that signal distal tissues to coordinate systemic metabolism. Fibroblast growth factor 21 (FGF21) and growth differentiation factor 15 (GDF15) are considered to be involved in mitochondrial hormesis, in that they have acutely beneficial effects in alleviating cardiometabolic disorders, but chronically elevated levels are linked to morbidity [4, 5]. FGF21 and GDF15 are reported to be useful biomarkers for MD diagnosis in non-HIV settings [6–8], with good discriminatory power in children at serum levels of 300pg/mL and 550pg/mL, respectively [2]. Furthermore, studies in adults living with HIV infection show associations of high FGF21 or GDF15 levels with lipodystrophy, insulin resistance, cardiovascular disease, and overall mortality [9, 10]. Thus, FGF21 and GDF15 are biomarkers for MD in non-HIV settings, and are indicators of cardiometabolic complications in adults living with HIV.

To our knowledge, no study has been published to date regarding the circulating values of these mitokines in CPHIV. This population is described to be at higher risk for mitochondrial dysfunction, likely as result of extended exposure to both HIV itself [11], and antiretroviral therapies (ART) used to contain infection [12]. Mitochondrial dysfunction is purported to be one major contributor to cardiometabolic complications, and studies involving CPHIV have described higher levels of inflammation and cardiovascular dysfunction compared to HIV-exposed but uninfected controls [13]. Furthermore, studies describe higher prevalence of IR compared with those reported for HIV-uninfected non-obese youth [14], lower copies of mtDNA in peripheral blood mononuclear cells (PBMCs) from those with IR versus without, and an inverse relationship between fasting glucose levels and OXPHOS Complex I enzymatic

activity [15]. High-throughput analyses of OXPHOS capacities within PBMCs also demonstrated lower basal and maximal respiratory levels in participants with IR compared to insulin-sensitive CPHIV [16]. Therefore, CPHIV may greatly benefit by identification of relatively-inexpensive, quickly-assayed, and minimally-invasive biomarkers of mitochondrial dysfunction, in hopes of alleviating progression to overt MD and/or other cardiometabolic complications.

We therefore sought to extend studies of these mitokines to include CPHIV, and hypothesized that circulating levels of FGF21 and/or GDF15 would be higher in children displaying "possible" MD (herein MDC score ≥3 points) than in those presenting no detectable MDC. Our data indicate these mitokines to be indicators of mitochondrial dysfunction in this population.

## Materials and methods

### Study design and population

We studied participants in the Pediatric AIDS Clinical Trials Group 219/219C (n = 2931) (https://clinicaltrials.gov/ct2/show/study/NCT00006304) who were enrolled between May 4, 1993 and November 3, 2004. Participants were classified by their highest clinical score on the Mitochondrial Disease Criteria (MDC) scale of Wolf and Smeitink [1]; a maximum of 4 points could be ascribed given that study protocols collected neither biochemical nor histological assessments of MD, as described previously [12]. Organ systems presenting symptoms at time of case diagnoses were also noted per MDC of Wolf and Smeitink. Individuals with MDC scores ≥3 points (n = 160 MDC3; n = 52 MDC4), were defined herein as having "possible" mitochondrial disease (Fig 1). Individuals were included in our study sample if they had a plasma specimen available on or up to 6 months after the date of their highest clinical MDC assessment (n = 71 MDC3; n = 23 MDC4). Participants displaying no MDC (MDC0; n = 23) and having plasma samples available within the above time-frame were randomly selected to be matched to MDC4 participants based upon the calendar date of case diagnosis to control for sample integrity. A complete case approach was used for missing data unless the proportion of missing data was greater than 10% for any MDC group, in which case a missing indicator was created. The parent or guardian of each participant provided written and informed consent for the child. This study was approved by the institutional review boards at the following institutions:

University of New Jersey Medical and Dental School—Department of Pediatrics, Division of Allergy, Immunology & Infectious Diseases, Boston Medical Center, Division of Pediatric Infectious Diseases, Med, Children's Hospital LA—Department of Pediatrics, Division of Clinical Immunology & Allergy, Long Beach Memorial Medical Center, Miller Children's Hospital, Harbor—UCLA Medical Center—Department of Pediatrics, Division of Infectious Diseases, Johns Hopkins Hospital & Health System—Department of Pediatrics, Division of Infectious Diseases, University of Maryland Medical Center, Division of Pediatric Immunology & Rheumatology, Texas Children's Hospital, Allergy & Immunology Clinic, Cook County Hospital, Children's Hospital of Columbus, Ohio, University of Miami Miller School of Medicine, Division of Pediatric Immunology & Infectious Disease, University of California San Francisco School of Medicine, Department of Pediatrics, Children's Hospital & Research Center Oakland, Pediatric Clinical Research Center & Research Lab, University of California San Diego Mother, Child & Adolescent HIV Program, Duke University School of Medicine—Department of Pediatrics, Children's Health Center, University of North Carolina at Chapel Hill School of Medicine—Department of Pediatrics, Division of Immunology and Infectious Diseases, Schneider Children's Hospital, Harlem Hospital Center, New York University School of

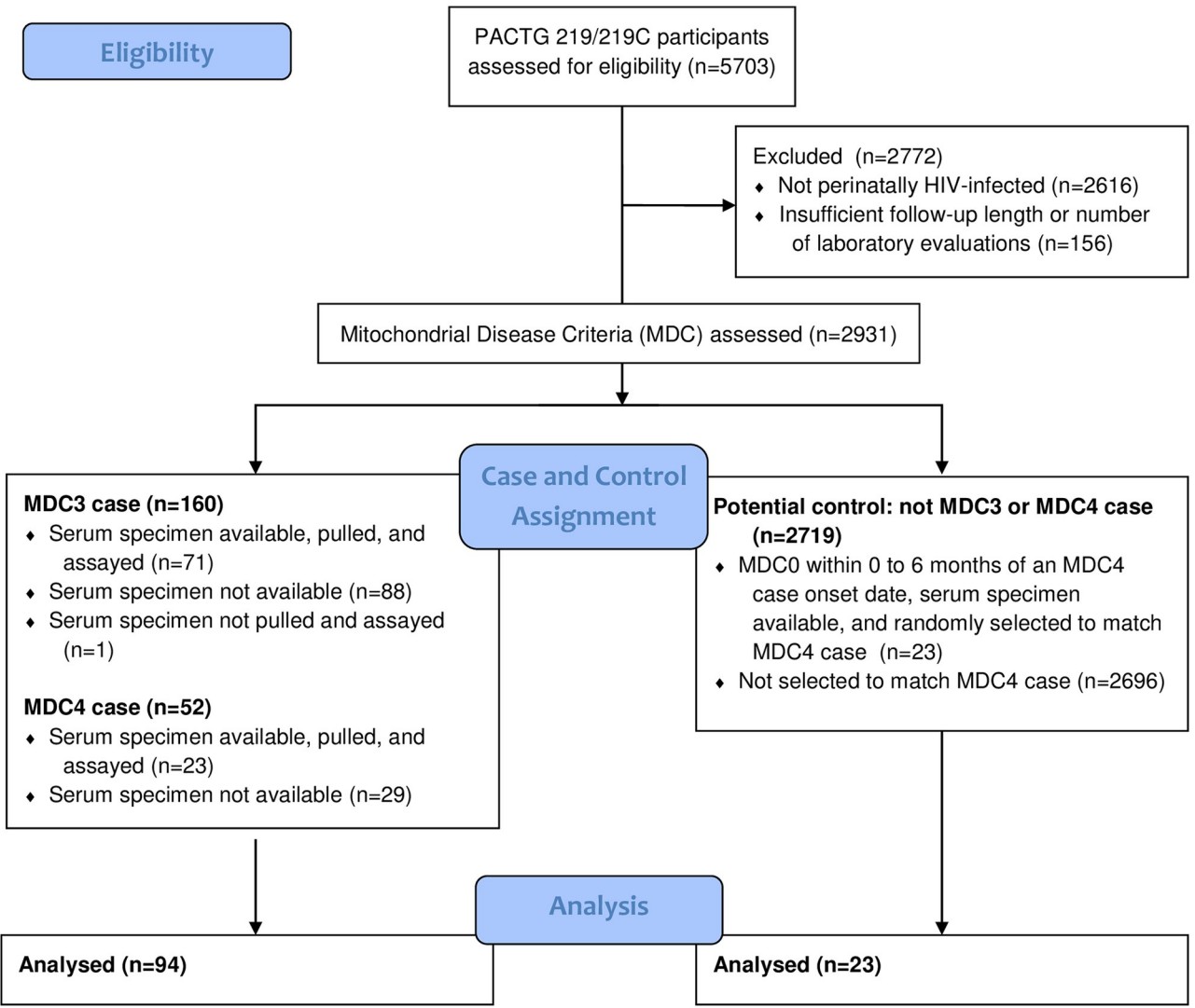

**Fig 1. Flow diagram of cross-sectional analysis of PACTG 219/219C cohort.**

Medicine, Division of Pediatric Infectious Diseases, Children's National Medical Center, ACT, University of Washington School of Medicine—Children's Hospital and Regional Medical Center, University of Illinois College of Medicine at Chicago, Department of Pediatrics, Yale University School of Medicine—Department of Pediatrics, Division of Infectious Disease, SUNY at Stony Brook School of Medicine, Division of Pediatric Infectious Diseases, Howard University Hospital, Department of Pediatrics & Child Health, LA County/University of Southern California Medical Center, University of Florida Health Science Center Jacksonville, Division of Pediatric Infectious Disease & Immunology, North Broward Hospital District, Children's Diagnostic & Treatment Center, University of Rochester Medical Center, Golisano Children's Hospital, Medical College of Virginia, St. Jude Children's Research Hospital, Department of Infectious Diseases, University of Puerto Rico, U. Children's Hospital AIDS, Children's Hospital of Philadelphia, Center for Pediatric & Adolescent AIDS, St. Christopher's Hospital for Children/Drexel University College of Medicine, Bronx-Lebanon Hospital Center, Infectious Diseases, New York Medical College/Metropolitan Hospital Center, University

of Massachusetts Memorial Children's Medical School, Department of Pediatrics, Baystate Health, Baystate Medical Center, Connecticut Children's Medical Center, Medical College of Georgia School of Medicine, Department of Pediatrics, Division of Infectious Disease, University of South Alabama College of Medicine, Southeast Pediatric ACTU, LSU Health Sciences Center, Tulane University Health Sciences Center, St. Josephs Hospital and Medical Center, Cooper University Hospital—Children's Hospital Boston, Division of Infectious Diseases, David Geffen School of Medicine at UCLA—Department of Pediatrics, Division of Infectious Diseases, Children's Hospital of Orange County, Children's Memorial Hospital—Department of Pediatrics, Division of Infectious Disease, University of Chicago—Department of Pediatrics, Division of Infectious Disease, Mt. Sinai Hospital Medical Center—Chicago, Women's & Children's HIV Program, Columbia University Medical Center, Pediatric ACTU, Incarnation Children's Center, Cornell University, Division of Pediatric Infectious Diseases & Immunology, University of Miami Miller School of Medicine—Jackson Memorial Hospital, Bellevue Hospital (Pediatric), San Francisco General (Pediatric), Phoenix Children's Hospital, Metropolitan Hospital Center (N.Y.), University of Cincinnati, SUNY Downstate Medical Center, Children's Hospital at Downstate, North Shore University Hospital, Jacobi Medical Center, University of South Florida—Department of Pediatrics, Division of Infectious Diseases, Cornell University, Oregon Health & Science University—Department of Pediatrics, Division of Infectious Diseases, Children's Hospital of the King's Daughters, Infectious Disease, Lincoln Medical & Mental Health Center, Mt. Sinai School of Medicine, Division of Pediatric Infectious Diseases, Emory University Hospital, San Juan City Hospital, UMDNJ—Robert Wood Johnson, Ramon Ruiz Arnau University Hospital, Medical University of South Carolina, SUNY Upstate Medical University, Department of Pediatrics, Wayne State University School of Medicine, Children's Hospital of Michigan, Children's Hospital at Albany Medical Center, Children's Medical Center of Dallas, Children's Hospital—University of Colorado at Denver and Health Sciences, Center, Pediatric Infectious Diseases, Columbus Children's Hospital, University of Florida College of Medicine—Department of Pediatrics, Division of Immunology, Infectious Diseases & Allergy, University of Mississippi Medical Center, Palm Beach County Health Department, Children's Hospital LA—Department of Pediatrics, Division of Adolescent Medicine, Vanderbilt University Medical Center, Division of Pediatric Infectious Diseases, Washington University School of Medicine at St. Louis, St. Louis Children's Hospital, Children's Hospital & Medical Center, Seattle ACTU, Oregon Health Sciences University, St. Luke's-Roosevelt Hospital Center, Montefiore Medical Center—Albert Einstein College of Medicine, Children's Hospital, Washington, D.C., Children's Hospital of the King's Daughters, University of Alabama at Birmingham, Department of Pediatrics, Division of Infectious Diseases, Columbus Regional HealthCare System, The Medical Center, Sacred Heart Children's Hospital/CMS of Florida, Bronx Municipal Hospital Center/Jacobi Medical Center.

## Clinical and laboratory measurements

Clinical and laboratory measurements taken closest to the date of the blood collection (within 6 months before or 1 month after) were used in analyses herein. These included routine anthropometric measurements [height, weight, body mass index (BMI), Tanner stage], virological and immunological measures [HIV viral RNA load, CD4+ T cell (CD4) counts], and liver/muscle aminotransferase marker levels. BMI z-scores adjusted for age and sex according to CDC growth standards were used for analyses. Clinical definitions of elevated aminotransferase markers were taken from medical records based on laboratory normals. Charts were reviewed for ART regimen history (past, current, and lifetime duration). ART regimens were classified based upon their inclusion of drugs previously described as associated with MD [12].

Regimens containing stavudine were compared to regimens containing zidovudine but not stavudine, and other regimens containing neither stavudine nor zidovudine. Clinical samples and measures were collected following standardized ACTG/IMPAACT Laboratory protocols from version 4.0 manual: https://www.hanc.info/labs/labresources/procedures/Pages/actgImpaactLabManual.aspx.

### Plasma assays

Blood was collected in acid citrate dextrose tubes, with plasma isolated and stored at -80C in biorepositories until shipped to the University of Hawaii on dry ice. ELISA kits from BioVendor (Asheville, North Carolina, USA) for FGF21 (cat. #RD191108200R) and GDF15 (cat. #RD191135200R) were run according to manufacturer's protocols, using duplicate wells per sample. Clinical cutoff values of each mitokine, taken from Montero et al. [2], were used for sensitivity/specificity analyses herein.

### Data analyses

Characteristics of MDC0, MDC3, and MDC4 participants, including sociodemographic data and HIV-specific characteristics, were summarized by the frequency (%) or median [inter-quartile ranges shown as quartiles (Q)1 and Q3]. Comparisons between the MDC4 and MDC0 groups were performed in matched analyses using paired Wilcoxon rank sum tests for continuous variables and McNemar's tests for categorical variables, while the unmatched comparisons of MDC3 and MDC0 for continuous or categorical variables used Wilcoxon rank sum or Fisher's Exact tests, respectively. Bowker's test of symmetry was also performed where appropriate. Natural logarithmic transformations were used for FGF21 and GDF15, to more closely approximate a normal distribution and improve robustness of analyses. Spearman correlation coefficients were calculated to assess associations between the mitokine concentrations overall and separately by MDC status. Conditional logistic regression models were fit to estimate the odds ratios and 95% Confidence Intervals (CI) for MDC4 vs MDC0 for each mitokine in a separate model, both unadjusted and adjusted for each potential confounder individually. To estimate the odds ratio of MDC3 vs MDC0 with each mitokine, separate logistic regression models were fit after adjustment for calendar year and age at the time of the case definition. The target sample size for primary comparisons of MDC4 vs MDC0 was n = 50, assuming a paired *t*-test with 5% Type I error rate and 76.5% power to detect an effect size of 0.4 standard deviations. Based on literature estimates for mean FGF21 concentration in children and standard deviation estimated based on normality, this sample size was powered to detect a difference between case and control of 96 pg/mL [17].

All analyses were performed using SAS Version 9.4 (Cary, NC, USA). The dataset used for this analysis is available by request on the PHACS website (https://phacsstudy.org/Our-Research/Data-Request-Form).

## Results

The final study population consisted of 117 CPHIV (MDC4 = 23, MDC3 = 71, MDC0 = 23). At the time of the MDC case definition, CPHIV with MDC4 were older than matched controls [median (Q1, Q3) age 12.3 (6.5, 14.7) vs 8.7 (5.6, 11.0) years, respectively] (Table 1). Distributions for sex, Tanner stage, age at ART initiation, BMI z-scores, or ART usage were similar between groups. However, the liver- and muscle-related enzymes, aspartate and alanine aminotransferases (AST and ALT, respectively), were elevated above expected ranges in a greater proportion of MDC4 cases than controls (35% of MDC4 vs 17% of MDC0 had elevated ALT; 48% of MDC4 vs 13% of MDC0 had elevated AST). The unmatched MDC3 group was similar

**Table 1. Association of clinical characteristics with MDC case status.**

| | Case status | | | p-value[*] | |
|---|---|---|---|---|---|
| | **MDC0** | **MDC3** | **MDC4** | **MDC3** | **MDC4** |
| **Participants** | | | | | |
| Number | 23 | 71 | 23 | | |
| **Age at time of case definition** | | | | | |
| Median (Q1, Q3) | 8.7 (5.6, 11.0) | 9.7 (6.4, 13.3) | 12.3 (6.5, 14.7) | 0.23 (a) | 0.032 (c) |
| **Age initiated ART** | | | | | |
| Median (Q1, Q3) | 1.7 (0.8, 6.6) | 2.1 (0.5, 6.4) | 1.6 (0.5, 7.6) | 0.81 (a) | 0.48 (c) |
| Missing | 1 (4%) | 0 (0%) | 0 (0%) | | |
| **ART regimen at specimen collection** | | | | | |
| Stavudine-containing | 12 (52%) | 35 (49%) | 9 (39%) | 0.64(b) | 0.42 (e) |
| Zidovudine-containing | 7 (30%) | 22 (31%) | 6 (26%) | | |
| Other | 2 (9%) | 14 (20%) | 8 (35%) | | |
| Missing | 2 (9%) | 0 (0%) | 0 (0%) | | |
| **Sex** | | | | | |
| Male | 10 (43%) | 39 (55%) | 14 (61%) | 0.47 (b) | 0.25 (d) |
| Female | 13 (57%) | 32 (45%) | 9 (39%) | | |
| **BMI z-score at specimen collection** | | | | | |
| Median (Q1, Q3) | 0.18 (-0.42, 0.92) | 0.15 (-0.54, 0.92) | -0.06 (-1.07, 0.92) | 0.50 (a) | 0.16 (c) |
| Missing | 2 (9%) | 6 (8%) | 5 (22%) | | |
| **Tanner stage** | | | | | |
| Tanner stage = 1 | 9 (39%) | 32 (45%) | 9 (39%) | 0.77 (b) | 0.66 (d) |
| Tanner stage = 2–5 | 6 (26%) | 18 (25%) | 11 (48%) | | |
| Missing | 8 (35%) | 21 (30%) | 3 (13%) | | |
| **ALT[#]** | | | | | |
| Within normal range | 18 (78%) | 54 (76%) | 15 (65%) | 0.77 (b) | 0.16 (d) |
| Elevated | 4 (17%) | 16 (23%) | 8 (35%) | | |
| Missing | 1 (4%) | 1 (1%) | 0 (0%) | | |
| **AST[#]** | | | | | |
| Within normal range | 19 (83%) | 48 (68%) | 12 (52%) | 0.17 (b) | 0.021 (d) |
| Elevated | 3 (13%) | 21 (30%) | 11 (48%) | | |
| Missing | 1 (4%) | 2 (3%) | 0 (0%) | | |
| **Viral load (copies/mL)** | | | | | |
| < 400 | 5 (22%) | 9 (13%) | 3 (13%) | 0.32 (b) | 0.66 (e) |
| 400–1000 | 6 (26%) | 12 (17%) | 4 (17%) | | |
| ≥ 1000 | 4 (17%) | 24 (34%) | 9 (39%) | | |
| Missing | 8 (35%) | 26 (37%) | 7 (30%) | | |
| **CD4 count (cells/mm$^3$)** | | | | | |
| ≤ 200 | 0 (0%) | 13 (18%) | 9 (39%) | 0.11 (b) | 0.006 (e) |
| 201–500 | 6 (26%) | 14 (20%) | 5 (22%) | | |
| ≥ 501 | 16 (70%) | 41 (58%) | 9 (39%) | | |

(*Continued*)

**Table 1.** (Continued)

| | Case status | | | p-value* | |
|---|---|---|---|---|---|
| | MDC0 | MDC3 | MDC4 | MDC3 | MDC4 |
| Missing | 1 (4%) | 3 (4%) | 0 (0%) | | |

*p-value displayed for comparison versus MDC0 controls; tests indicated by letter:

(a) Wilcoxon Rank Sum Test

(b) Fisher's Exact Test

(c) Paired Wilcoxon Rank Sum Test

(d) McNemar's Test

(e) Bowker's Test of Symmetry

#Classification of ALT and AST as elevated or within normal range based on age- and sex-specific reference values provided by laboratory or from Harriet Lane Handbook, 21st ed.

-ALT = alanine aminotransferase; ART = antiretroviral therapy; AST = aspartate aminotransferase; BMI = body mass index; CD4 = CD4+ T cells;

MDC = mitochondrial disease criteria; Q1 = quartile 1; Q3 = quartile 3.

to MDC0 controls for most parameters, though 30% of MDC3 cases had elevated AST compared to only 13% of MDC0 controls. HIV viral loads were similar between groups; however, a large proportion of each group lacked these data points within the specified window surrounding blood collection, and higher proportions of participants with "possible" MD had loads greater than 1000 copies/mL compared to controls (39% of MDC4, 34% of MDC3, and 17% of MDC0). A substantially larger proportion of MDC4 cases displayed CD4 counts below 200 cells/mm3 versus controls (39% vs 0%), and 18% of MDC3 participants also fell in this category.

Median concentrations of FGF21 (Fig 2a) and GDF15 (Fig 2b) were highest in MDC4 cases [143.9 (84.2, 502.3) and 1441.1 (436.7, 2514.4) pg/mL, respectively], next highest in MDC3 cases [104.0 (84.1, 167.9) and 726.5 (419.2, 2066.9) pg/mL], and lowest in MDC0 controls [89.4 (59.3, 130.8) and 484.7 (414.8, 790.3) pg/mL]. Distributions of both FGF21 (paired Wilcoxon rank sum p = 0.002) and GDF15 (paired Wilcoxon rank sum p<0.001) differed in MDC4 cases compared to matched MDC0 controls, while unmatched MDC3 cases differed from MDC0 values for FGF21 (Wilcoxon rank sum p = 0.022), but not for GDF15 (Wilcoxon rank sum p = 0.128).

Independent of case status, visual trends regarding distributions of FGF21 and GDF15 relating to other covariates were observed (data not shown). Participants on stavudine-containing regimens had higher median values of FGF21 compared to zidovudine-containing regimens [136 (85, 230) vs 89 (83, 139) pg/mL, respectively], but lower values of GDF15 [1600 (893, 2501) vs 529 (415, 1119) pg/mL]. The median values of FGF21 and GDF15, respectively, were highest in those who initiated ART between 6–10 years of age [135 (84, 148); 986 (532, 2065) pg/mL] compared to 0–1.9 yrs [111 (84, 150); 718 (429, 1603) pg/mL], 2–5 yrs [90 (84, 144); 447 (404, 1958) pg/mL] and 11+ yrs [112 (83, 421); 833 (476, 1907) pg/mL]. No clinically meaningful differences by Tanner stage or BMI were observed; however, for 27% and 11% of total participants, respectively, there were not data available within the window. Individuals with elevated compared to normal ALT displayed higher values of FGF21 [140 (90, 462) vs 90 (84, 141) pg/mL], and GDF15 [2281 (790, 2979) vs 635 (396, 1441) pg/mL]. The same was observed for elevated versus normal AST regarding FGF21 [141 (84, 421) vs 90 (84, 140) pg/mL] and GDF15 [1446 (718, 2901) vs 533 (386, 1441) pg/mL]. Distributions of mitokine values were not different based on gender or viral load.

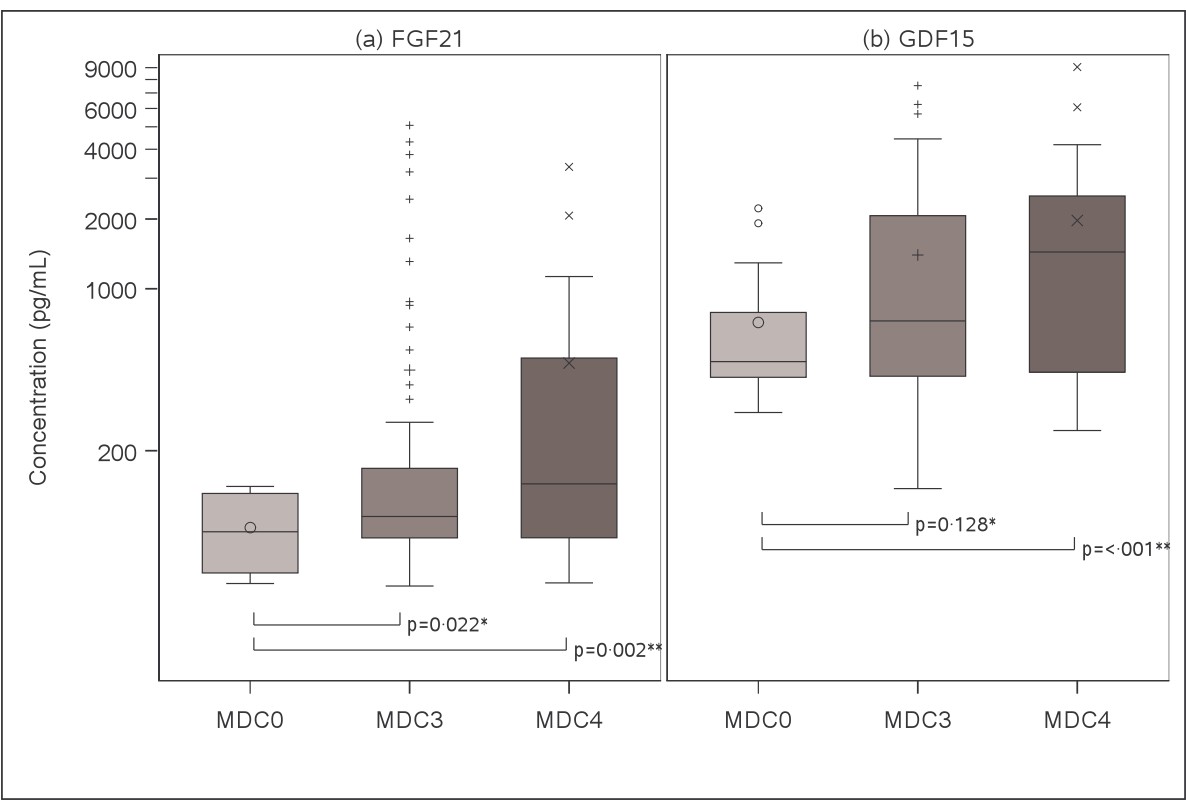

**Fig 2. Plasma mitokine concentrations based on case status.** (a) Distributions of FGF21 values are shown across MD cases and controls as determined in plasma from the timepoint closest to MDC case definition. (b) GDF15 distributions across "possible" MD cases and controls from same plasma timepoint as in (a). Plasma values of each mitokine determined in duplicate via ELISA. Bottom and top box edges represent quartiles one and three, respectively, while horizontal line within box indicates group median. Vertical error bars indicate one and a half times the interquartile range, with individual observations outside that range displayed as small icons (circles, crosses, and X's). The largest icon for each group indicates the mean value. * *Wilcoxon rank sum test comparing MDC3 to MDC0 controls.* ** *Paired Wilcoxon rank sum test comparing MDC4 to MDC0 controls.* FGF21 = fibroblast growth factor 21; GDF15 = growth differentiation factor 15; MDC = mitochondrial disease criteria.

Of the MDC3 cases, 4 (6%) exhibited muscular presentation, 33 (46%) exhibited central nervous system (CNS) presentation, and 34 (48%) presented with an abnormality in another organ system (data not shown). Of the MDC4 cases, 14 (61%) exhibited CNS presentation and the remaining 9 (39%) presented with involvement of another organ system (data not shown). Participants presenting with muscular system effects at diagnosis had the highest median values for FGF21 [317 (89, 614) pg/mL], whereas median values of GDF15 [1495 (455, 2361) pg/mL] were highest in those participants displaying MD in multiple organ systems.

Across all participants or within groups, mild to moderate correlations were observed between the two mitokines, and between the mitokines and current CD4 counts, while correlations were negligible for mitokine values with current viral load. Specifically, across all groups combined, the correlation between FGF21 and GDF15 concentrations was 0.33 (p<0.001; data not shown), while Fig 3 displays correlational analyses within individual groups. Correlations between these two mitokines were -0.10 for MDC0 (Fig 3a), 0.38 for MDC3 (Fig 3b) and 0.19 for MDC4 (Fig 3c). Current CD4 count negatively correlated with FGF21 (r = -0.21; p = 0.028) and GDF15 (r = -0.27; p = 0.005) across all participants (data not shown); and within MDC4 cases (data not shown), CD4 count also negatively correlated with FGF21 (r = -0.31; p = 0.151), and GDF15 (r = -0.43; p = 0.043).

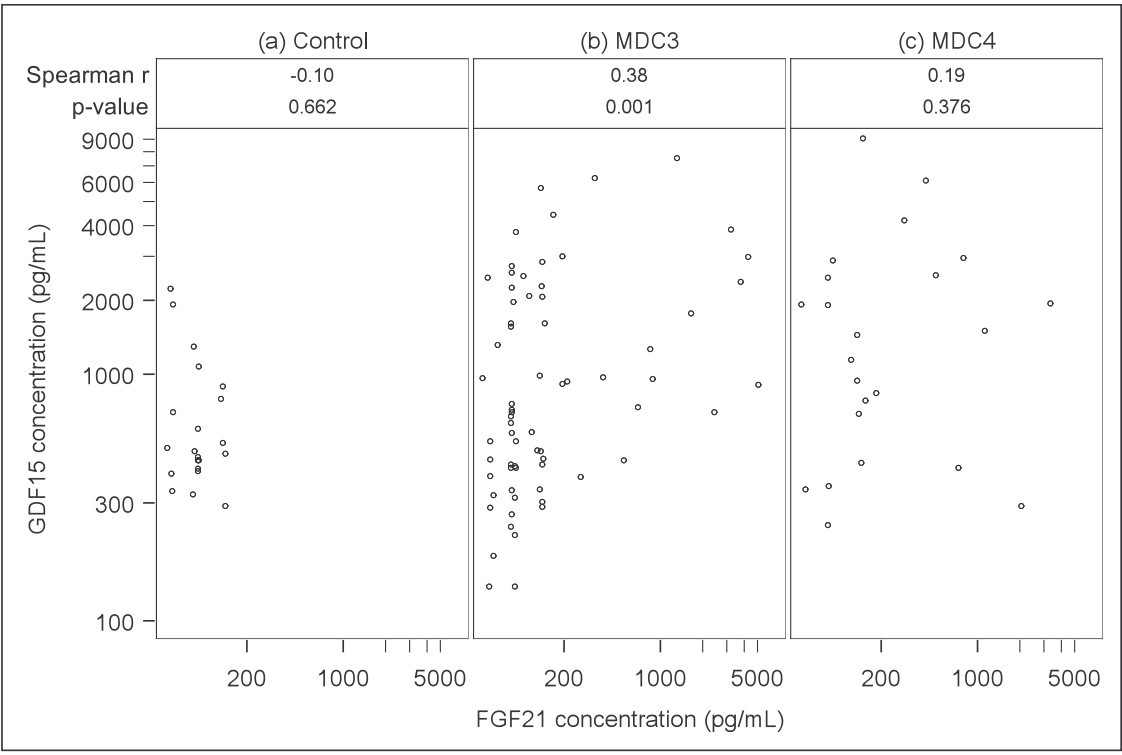

**Fig 3. Correlational analyses of FGF21 and GDF15 concentrations by case status.** Associations between plasma values for FGF21 and GDF15 are shown for (a) MDC0 controls, (b) MDC3 cases, and (c) MDC4 cases. Spearman's r coefficients and associated p values for each case status group shown above. FGF21 = fibroblast growth factor 21; GDF15 = growth differentiation factor 15; MDC = mitochondrial disease criteria.

Considering just MDC4 cases, the previously published cutoff value for FGF21 of 300pg/ml displayed a sensitivity of 30.4% (Table 2). The overall sensitivity of the cutoff value for FGF21 alone was 21.3%, correctly classifying only 20 of the 94 combined MDC3 and MDC4 cases. The specificity for FGF21 was 100%, correctly classifying all of the 23 controls. The GDF15 cutoff of 550 pg/mL correctly classified 17 (73.9%) of the 23 MDC4 cases, and 58 (61.7%) of the 94 combined MDC3 and MDC4 cases. The specificity of the GDF15 cutoff was 56.5%.

**Table 2. Individual discrimination of published mitokine clinical cutoffs by case status.**

| Clinical cutoff classification[#] | MDC case status | | | |
|---|---|---|---|---|
| | MDC0[*] | MDC3[*] | MDC4 | MDC3/4 |
| Participants (n) | 23 | 71 | 23 | 94 |
| FGF21 ≥ 300 pg/mL | 0 (0.0%) | 13 (18.3%) | 7 (30.4%) | 20 (21.3%) |
| FGF21 < 300 pg/mL | 23 (100.0%) | 58 (81.7%) | 16 (69.6%) | 74 (78.7%) |
| GDF15 ≥ 550 pg/mL | 8 (34.8%) | 41 (57.7%) | 17 (73.9%) | 58 (61.7%) |
| GDF15 < 550 pg/mL | 13 (56.5%) | 27 (38.0%) | 6 (26.1%) | 33 (35.1%) |
| FGF21 ≥ 300 & GDF15 ≥ 550 pg/mL | 0 (0.0%) | 12 (16.9%) | 5 (21.7%) | 17 (18.1%) |

[#]Cutoffs published by Montero et al.

[*]2 controls and 3 MDC3 cases missing GDF15 concentration.

FGF21 = fibroblast growth factor 21; GDF15 = growth differentiation factor 15; MDC = mitochondrial disease criteria.

When cutoff values were combined, no MDC0 controls had both mitokine concentrations above their respective cutoffs. However, only 12 (17%) of MDC3 cases and 5 (22%) of MDC4 cases had concentrations of both mitokines above the cutoffs, with a joint sensitivity of 18% among the 94 combined MDC3 and MDC4 cases.

Results of the conditional logistic regression models assessing the association of the natural log-transformed mitokine values with the odds of being diagnosed as a MDC4 case compared to a matched control are shown in Fig 4. The unadjusted odds of being a MDC4 case was 5.25 times higher than being a control with each one natural log unit increase of FGF21 [95% CI 1.06–25.92; p = 0.042] and 3.49 times higher for each one natural log unit increase of GDF15 (95%CI 1.19–10.25; p = 0.023). The results were generally similar when models were adjusted for one covariate at a time, although the confidence intervals for FGF21 were much wider in adjusted models. The adjusted odds of being a MDC3 case compared to control (data not shown) were 3.65 times higher with each one natural log unit increase of FGF21 (95%CI 1.15 11.58; p = 0.03) and 1.69 times higher for each one natural log increase of GDF15 (95%CI 0.91 3.13; p = 0.10).

## Discussion

In our cross-sectional analysis of CPHIV who did or did not meet clinical definitions of "possible" mitochondrial disease, we found significantly higher plasma levels of the mitokines FGF21 and/or GDF15 in those with MDC score $\geq$3. Across all participants, a correlation between the two markers was observed, as highlighted within the MDC3 group. Higher levels of either mitokine were associated with a greater likelihood of being diagnosed with "possible" MD. To our knowledge, this is the first study to assess the relationship of these biomarkers with mitochondrial dysfunction in CPHIV.

Our findings are consistent with other studies in non-HIV populations demonstrating the overall utility of FGF21 and GDF15 as diagnostic biomarkers of MD [2, 3, 6–8], though some studies have raised concerns regarding their discriminatory powers against other non-mitochondrial disorders [3, 18, 19]. In the general population, these markers have also been shown to be associated with metabolic disorders, cardiovascular outcomes, and mortality [4, 5], and evidence in HIV populations suggest similar connections [10]. While our study objective was not to assess the relationship between mitokines and cardiometabolic outcomes, this may be a potential area of study given that CPHIV may have greater cardiometabolic risk than their uninfected peers [20].

HIV exposure or infection may itself directly affect mitokine levels, as described for FGF21 [9], and therefore the reasoning for other factors correlating with mitokine levels need to be carefully considered within the CPHIV population. In contrast to some studies, we found no significant associations between levels of these mitokines with sex [21], antiretroviral class usage [22], or viral load [23]; however, we did note negative correlations between current CD4 counts with FGF21 and GDF15 values, especially within the MDC4 group, potentially further linking mitokines to composition of immune cells post-HIV infection with metabolic dysfunction, as suggested previously for FGF21 [19]. The association of nadir CD4 count with future insulin resistance and mitochondrial dysfunction has been observed in CPHIV [24]. Indeed, a large field of work has evolved describing the impact that HIV and ART have upon the mitochondrion, and how that shapes the composition of the immune system, with the subsequent proinflammatory signals in susceptible individuals potentially leading to systemic metabolic disorders being a plausible hypothesis. Along these lines, we observed that a greater proportion of MD cases displayed viral loads above 1000 copies/mL; however, the lack of data within the window surrounding blood collection precluded further assessments in our study or

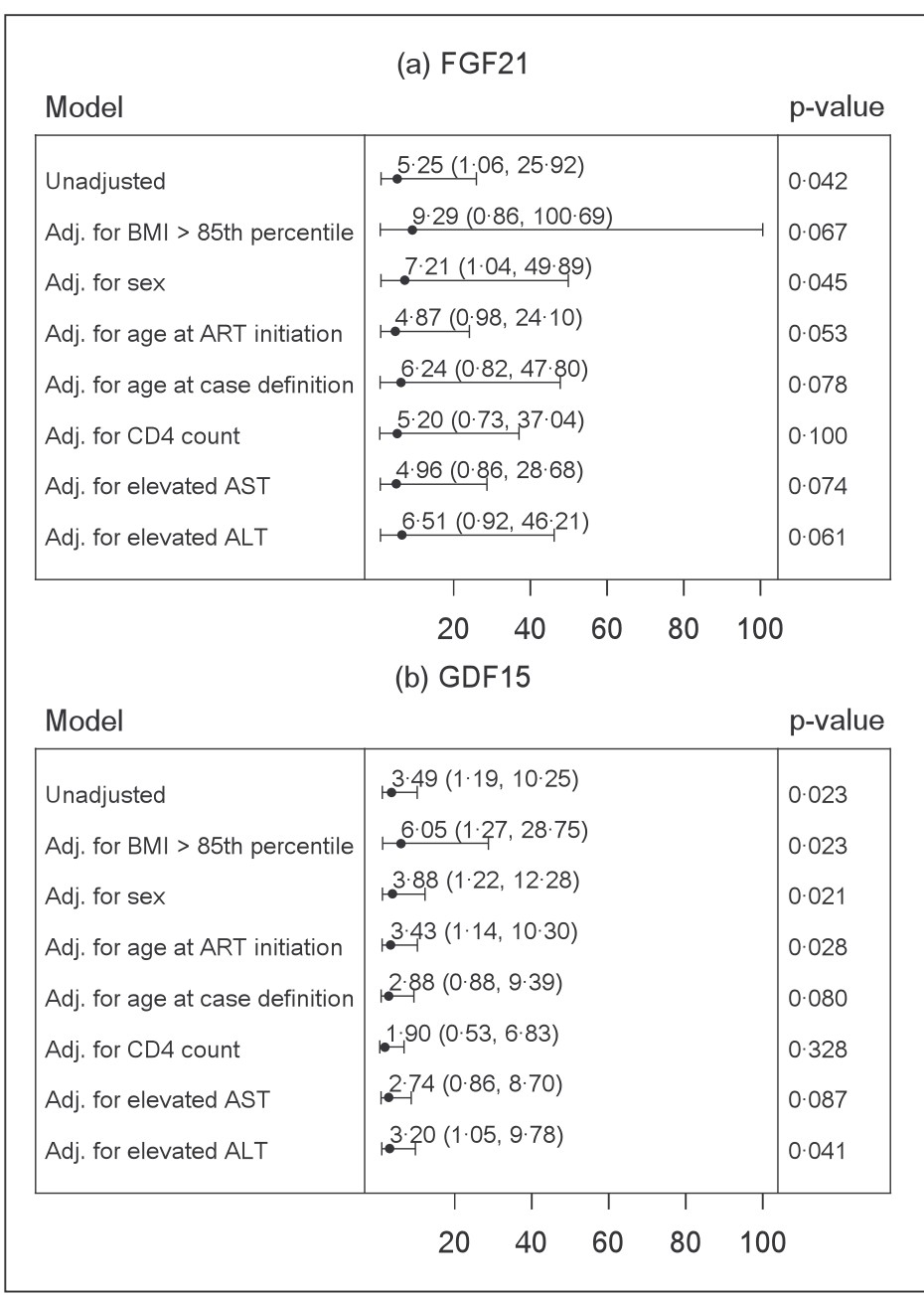

**Fig 4. Bivariate conditional logistic regression model results.** (a) Models were created comparing natural log-transformed values of FGF21 for MDC4 cases versus MDC0 controls, with individual adjustments for covariates shown. (b) Similar models were created for GDF15 values. Estimates for the odds of MDC4 status vs. MDC0 status obtained from the indicated model (left panels) are represented as points, with bars indicating 95% CI. Statistical p-values for the odds ratio are shown in right panels. Adj. = adjusted; ALT = alanine aminotransferase; ART = antiretroviral therapy; AST = aspartate aminotransferase; BMI = body mass index; CI = confidence interval; FGF21 = fibroblast growth factor 21; GDF15 = growth differentiation factor 15; MDC = mitochondrial disease criteria.

comparisons to a recent report [23]. Potential effects relating to stavudine usage upon FGF21 and zidovudine usage upon GDF15 were noted, highlighting the possible impact that specific ART regimens may have upon these mitokines and mitochondrial function. While these regimens are being used less-commonly and are being replaced by newer, potentially less mito-

toxic drugs, questions still remain regarding whether expression of these mitokines is being driven by the virus itself or by newer therapeutic regimens as well.

Intriguingly, we observed somewhat higher mitokine values in participants who initiated ART regimens later in life, as well as in those who were diagnosed with "possible" MD at older ages; although, no associations with ART duration or Tanner stage were found with either mitokine, and no difference was observed between matched groups based on this metric of aging. However, advanced aging has been linked to high mitokine expression in other studies [4], and our diagnosis-date matched, peripubertal groups did show differences in their ages at the time of the case definition. The limited available data published, especially as relating to CPHIV, suggest no effect of age upon concentrations of FGF21 or GDF15 over the peripubertal time period when accounting for anemia, diabetes, obesity, or cardiovascular disease [17, 25–27]. Our matched groups differed slightly in BMI z-scores, with the MDC4 cases displaying lower scores than controls, which may have impacted GDF15 values [28], though most studies suggest these mitokines generally associate with higher BMI. We also observed that elevated levels of ALT and AST were associated with higher mitokine concentrations, and a greater proportion of cases had elevated levels compared to controls. These markers have been linked previously to FGF21 levels during hepatitis B infection in non-HIV settings, and hepatic steatosis in HIV settings [29, 30]. Other reports suggest that starvation, bone marrow density, and muscle wasting may also alter ALT/AST and/or FGF21/GDF15 levels, thereby further highlighting the confounding clinical presentations that diagnosing clinicians often face, as well as the need to determine mechanisms and order of presentation of these biomarkers.

Strengths of our study include the use of calendar-date matched groups to control for potential sample integrity loss over the decade of patient enrollment, as well as the inclusion of patients with MDC3 scores to assess the robustness of our hypothesis at lower limits of then-current diagnostic practice. Our study was limited by our inability to more rigidly adhere to the criteria of Wolf and Smeitink due to study protocol limitations, as we could not ascribe "probable" or "definite" diagnoses of MD, thereby potentially mischaracterizing the severity of disease in some participants. Newer DNA-sequencing technologies evaluating both genomic and mitochondrial sequences could be informative. Lastly, we were not able to adjust for the degree/presence of insulin resistance, and children perinatally-exposed to HIV may be more susceptible to this outcome [15]. FGF21 has been linked to insulin resistance and gender differences, potentially driven by elevated circulating triglyceride levels [21], which have been described in CPHIV as well [16].

Studies in the future would benefit from longitudinal assessment of larger cohorts, especially those including peripubertal participants due to physiological changes in insulin sensitivity over this period [14]. When assessing the presenting organ system at diagnosis, we observed higher FGF21 values in those with muscular sequelae; however, our limited sample size precluded statistical analysis. Yet, this observation agrees with FGF21 being related to muscular system presentation [3, 6], while GDF15 may be linked to more generalized mitochondrial disease and indicative of disease severity and/or progression [31]. Our analyses using mitokine cutoff values previously established in non-HIV MD settings indicate that future studies could define clinically meaningful diagnostic algorithms for screening the CPHIV population, as FGF21 had excellent specificity. Montero et al. also described correlations of aminotransferase levels with mitokine expression similar to what was found herein, indicating future studies could perform more stringent comparisons of these four biomarkers in CPHIV as has been done in the non-HIV setting [7, 31]. Lastly, studies assessing efficacy of intervention strategies aimed at mitigating cardiometabolic complications in CPHIV may benefit by monitoring these mitokines, as suggested in a study of adults living with HIV infection that determined associations of FGF21 expression with the beneficial "beiging" of adipose tissues and better energy homeostasis in response to lifestyle modification [32].

Overall, we suggest that addition of simple, inexpensive assays to quantify FGF21 and GDF15 levels during routine blood monitoring may greatly assist in identifying those CPHIV who are at higher risk of developing mitochondrial dysfunction-linked cardiometabolic complications, above-and-beyond the risks already associated with HIV infection and ART.

## Supporting information

**S1 Checklist. STROBE statement—Checklist of items that should be included in reports of observational studies.**
(DOC)

## Acknowledgments

We kindly point our readers to our statement of acknowledgments here: https://www.phacsstudy.org/About-Us/219_219C.Acknowledgements.

Note: The conclusions and opinions expressed in this article are those of the authors and do not necessarily reflect those of the National Institutes of Health or U.S. Department of Health and Human Services.

## Author Contributions

**Conceptualization:** Marilyn Crain, Mariana Gerschenson.

**Data curation:** Denise L. Jacobson, Carly Broadwell, Brad Karalius.

**Formal analysis:** Denise L. Jacobson, Carly Broadwell, Brad Karalius.

**Funding acquisition:** Mariana Gerschenson.

**Investigation:** Greg S. Gojanovich.

**Project administration:** Marilyn Crain, Mariana Gerschenson.

**Resources:** Brian Kirmse, Mitchell E. Geffner, Jennifer Jao, Russell B. Van Dyke, Elizabeth J. McFarland, Margarita Silio.

**Visualization:** Denise L. Jacobson, Carly Broadwell, Brad Karalius.

**Writing – original draft:** Greg S. Gojanovich.

**Writing – review & editing:** Greg S. Gojanovich, Brian Kirmse, Mitchell E. Geffner, Jennifer Jao, Russell B. Van Dyke, Elizabeth J. McFarland, Margarita Silio, Marilyn Crain, Mariana Gerschenson.

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
