## [Decision Letter · Decision Letter 0]

24 Sep 2021

PONE-D-21-22547Associations of FGF21 and GDF15 with mitochondrial dysfunction in children living with perinatally-acquired HIV: a cross-sectional evaluation of Pediatric AIDS Clinical Trials Group 219/219CPLOS ONE

Dear Dr. Gerschenson,

Thank you for submitting your manuscript to PLOS ONE. After careful consideration, we feel that it has merit but does not fully meet PLOS ONE’s publication criteria as it currently stands. Therefore, we invite you to submit a revised version of the manuscript that addresses the points raised during the review process.

We look forward to receiving your revised manuscript.

Kind regards,

Marianne Clemence, Associate Editor, PLOS ONE, on behalf of

Qigui Yu, M.D./Ph.D

Academic Editor

PLOS ONE

Journal Requirements:

Additional Editor Comments (if provided):

Reviewers' comments:

Reviewer's Responses to Questions

**Comments to the Author**

1. Is the manuscript technically sound, and do the data support the conclusions?

Reviewer #1: Yes

Reviewer #2: Yes

Reviewer #3: Partly

2. Has the statistical analysis been performed appropriately and rigorously? 

Reviewer #1: Yes

Reviewer #2: Yes

Reviewer #3: Yes

3. Have the authors made all data underlying the findings in their manuscript fully available?

Reviewer #1: Yes

Reviewer #2: Yes

Reviewer #3: No

4. Is the manuscript presented in an intelligible fashion and written in standard English?

Reviewer #1: Yes

Reviewer #2: Yes

Reviewer #3: Yes

5. Review Comments to the Author

Reviewer #1: This study looked at the relationship between two biomarkers, FGF21 and GDF15, and their relationship to Mitochondrial Disease in CPHIV. Metabolic and mitochondrial pathology are hallmarks of HIV infection due to its’ propensity for manipulating host metabolism via viral proteins. Mitokines, such as those in this study, are produced as a result of mitochondrial stress and can therefore indication HIV driven mitochondrial dysfunction.

The authors were able to find increased trends of both mitokines with increasing severity of disease, however, they did not find correlations with viral load (which should be highlighted more in the study). Furthermore, their usage of “significant correlation” is not correct as an R of greater than 0.5 would have been required. All correlations found in this study (other than control FGF21 v. GDF15) were minor to moderate. Samples from individuals with confirmed mitochondrial disease would have been helpful. The addition of mitochondrial dysfunction measurements in each of the groups would also have been helpful. Despite this, the paper does sufficiently show evidence for their claims.

Reviewer #2: It’s a well-designed study, the statistical methods used are appropriate. I have no concerns about this study except one question: the mitochondrial diseases in most cases are inherited, do you have data in related to the CPHIV parents’ mitochondrial conditions and tried to analyze the association of mitokines between parents and children?

Reviewer #3: A clinical trial was conducted which aimed to assess the relationship between FGF21 and GDF15 levels in children with mitochondrial disease (MD) and HIV. A matched pairs analysis was conducted for those with MDC of 4. Unmatched cases with MDC of 3 were also assessed. FGF21 and GDF15 were highest in MDC4, moderate in MDC3 and lowest in controls.

Minor revisions:

1- At each mention of the Wilcoxon test, specify the exact type.

2- The estimate for a correlation coefficient is represented by r instead of rho.

3- The p-value associated with a correlation is a test of the null hypothesis: correlation equal to zero; however, the absolute magnitude of the coefficient indicates the strength of the linear relationship between two variables. In general, the strength or correlation coefficient is the more important statistic to focus on.

Below is a table for interpreting correlation coefficients:

Coefficient (absolute value) Interpretation

0.90 - 1.0 Very Strong

0.70 - 0.89 Strong

0.40 - 0.69 Moderate

0.10 - 0.39 Weak

less than 0.10 Negligible correlation

4- Indicate the date range subjects were enrolled in the study.

5- Cite the statistical software used for the analysis.

6- State and justify the study’s target sample size with a pre-study statistical power calculation.

The power calculation should include: (1) the estimated outcomes in each group; (2) the α (type I) error level; (3) the statistical power (or the β (type II) error level); (4) the target sample size and (5) for continuous outcomes, the standard deviation of the measurements.

6. PLOS authors have the option to publish the peer review history of their article (what does this mean?). If published, this will include your full peer review and any attached files.

Reviewer #1: No

Reviewer #2: No

Reviewer #3: No

---

## [Author Response · Author response to Decision Letter 0]

12 Nov 2021

Response to Reviewer Comments

“Associations of FGF21 and GDF15 with mitochondrial dysfunction in children living with perinatally-acquired HIV: a cross-sectional evaluation of Pediatric AIDS Clinical Trials Group 219/219C "

We would like to thank the editor and reviewers for their time and expertise with reviewing our manuscript. We agree that the suggested revisions make for a more robust and accurate manuscript, and have adapted it accordingly. We appreciate the reviewers stating that our manuscript is ‘a well-designed study, the statistical methods used are appropriate’ and that ‘the paper does sufficiently show evidence for their claims.’

Please see responses to specific reviewer remarks below in italics, and track changes have been used to highlight alterations within the manuscript, while mention of specific lines refers to this track-changed version. An unmarked, updated version without track changes is also supplied.

Editor’s Comments to the Authors 

Authors’ Response: We have found no retraction notices for any articles cited, however corrections were published to amend minor concerns in Montero et al. and Tsygankova et al., which did not result in meaningful differences to conclusions drawn upon herein. Additionally, a formatting error for reference notation within the manuscript was corrected on line 218.

Authors’ Response: We have amended our wording to indicate that we obtained the participant’s guardians consent on line 121.

Reviewers’ Comments to the Authors: 

Reviewer #1: This study looked at the relationship between two biomarkers, FGF21 and GDF15, and their relationship to Mitochondrial Disease in CPHIV. Metabolic and mitochondrial pathology are hallmarks of HIV infection due to its’ propensity for manipulating host metabolism via viral proteins. Mitokines, such as those in this study, are produced as a result of mitochondrial stress and can therefore indication HIV driven mitochondrial dysfunction.

The authors were able to find increased trends of both mitokines with increasing severity of disease, however, they did not find correlations with viral load (which should be highlighted more in the study). Furthermore, their usage of “significant correlation” is not correct as an R of greater than 0.5 would have been required. All correlations found in this study (other than control FGF21 v. GDF15) were minor to moderate. Samples from individuals with confirmed mitochondrial disease would have been helpful. The addition of mitochondrial dysfunction measurements in each of the groups would also have been helpful. Despite this, the paper does sufficiently show evidence for their claims.

Authors’ Response: We agree with the reviewer’s concern regarding significance statements relating to the correlations and have adjusted phrasing accordingly, with a specific change found on line 430 in the discussion.

We also agree that having samples from individuals with confirmed mitochondrial disease would have bolstered our data here, however this proposed sample collection was outside of the scope of our study, and the subjects within our study were not consented for histological/biochemical analyses that would have permitted clinical confirmation.

Reviewer #2: It’s a well-designed study, the statistical methods used are appropriate. I have no concerns about this study except one question: the mitochondrial diseases in most cases are inherited, do you have data in related to the CPHIV parents’ mitochondrial conditions and tried to analyze the association of mitokines between parents and children?

Authors’ Response: The authors completely agree that this line of study is warranted and, in fact, have already proposed such studies be performed in mother/child pairs, but are awaiting funding for this proposal.

Reviewer #3: A clinical trial was conducted which aimed to assess the relationship between FGF21 and GDF15 levels in children with mitochondrial disease (MD) and HIV. A matched pairs analysis was conducted for those with MDC of 4. Unmatched cases with MDC of 3 were also assessed. FGF21 and GDF15 were highest in MDC4, moderate in MDC3 and lowest in controls.

Minor revisions:

1- At each mention of the Wilcoxon test, specify the exact type.

2- The estimate for a correlation coefficient is represented by r instead of rho.

3- The p-value associated with a correlation is a test of the null hypothesis: correlation equal to zero; however, the absolute magnitude of the coefficient indicates the strength of the linear relationship between two variables. In general, the strength or correlation coefficient is the more important statistic to focus on.

Below is a table for interpreting correlation coefficients:

Coefficient (absolute value) Interpretation

0.90 - 1.0 Very Strong

0.70 - 0.89 Strong

0.40 - 0.69 Moderate

0.10 - 0.39 Weak

less than 0.10 Negligible correlation

4- Indicate the date range subjects were enrolled in the study.

5- Cite the statistical software used for the analysis.

6- State and justify the study’s target sample size with a pre-study statistical power calculation.

The power calculation should include: (1) the estimated outcomes in each group; (2) the α (type I) error level; (3) the statistical power (or the β (type II) error level); (4) the target sample size and (5) for continuous outcomes, the standard deviation of the measurements.

Authors’ Response:

1. We agree and have adjusted accordingly throughout the manuscript.

2. We have changed all instances of rho to r.

3. We have changed wording selection to align with these suggestions throughout the manuscript.

4. We have moved this manuscript section from the results to the methods so as to be more similar to commonly used formatting.

5. Thank you for catching this oversight; this can now be found on line 255.

6. A statement regarding power calculations can now be found starting on line 236.

---

## [Editor Report · Decision Letter 1]

6 Dec 2021

Associations of FGF21 and GDF15 with mitochondrial dysfunction in children living with perinatally-acquired HIV: a cross-sectional evaluation of Pediatric AIDS Clinical Trials Group 219/219C

PONE-D-21-22547R1

Dear Dr. Gerschenson,,

We’re pleased to inform you that your manuscript has been judged scientifically suitable for publication and will be formally accepted for publication once it meets all outstanding technical requirements.

Kind regards,

Qigui Yu, M.D./Ph.D

Academic Editor

PLOS ONE

---

## [Editor Report · Acceptance letter]

10 Dec 2021

PONE-D-21-22547R1 

Associations of FGF21 and GDF15 with mitochondrial dysfunction in children living with perinatally-acquired HIV: a cross-sectional evaluation of Pediatric AIDS Clinical Trials Group 219/219C 

Dear Dr. Gerschenson:

I'm pleased to inform you that your manuscript has been deemed suitable for publication in PLOS ONE. Congratulations! Your manuscript is now with our production department. 

Kind regards, 

on behalf of

Dr. Qigui Yu 

Academic Editor

PLOS ONE